# Uranium: The Nuclear Fuel Cycle and Beyond

**DOI:** 10.3390/ijms23094655

**Published:** 2022-04-22

**Authors:** Bárbara Maria Teixeira Costa Peluzo, Elfi Kraka

**Affiliations:** Computational and Theoretical Chemistry Group (CATCO), Department of Chemistry, Southern Methodist University, 3215 Daniel Ave, Dallas, TX 75275-0314, USA; bmpeluzo@smu.edu

**Keywords:** uranium, nuclear energy, uranium and health, uranium coordination chemistry, uranium nitrides, uranium triple bonds, local vibrational mode analysis, bond strength

## Abstract

This review summarizes the recent developments regarding the use of uranium as nuclear fuel, including recycling and health aspects, elucidated from a chemical point of view, i.e., emphasizing the rich uranium coordination chemistry, which has also raised interest in using uranium compounds in synthesis and catalysis. A number of novel uranium coordination features are addressed, such the emerging number of U(II) complexes and uranium nitride complexes as a promising class of materials for more efficient and safer nuclear fuels. The current discussion about uranium triple bonds is addressed by quantum chemical investigations using local vibrational mode force constants as quantitative bond strength descriptors based on vibrational spectroscopy. The local mode analysis of selected uranium nitrides, N≡U≡N, U≡N, N≡U=NH and N≡U=O, could confirm and quantify, for the first time, that these molecules exhibit a UN triple bond as hypothesized in the literature. We hope that this review will inspire the community interested in uranium chemistry and will serve as an incubator for fruitful collaborations between theory and experimentation in exploring the wealth of uranium chemistry.

## 1. Introduction

Uranium was discovered in 1789 by the German chemist Martin Heinrich Klaproth [1]. It is the most known and used actinide element mainly because of its usage in nuclear fuel processing; however, the application potential of uranium compounds is much broader, stretching, e.g., into the field of organometallic synthesis, catalysis and beyond, as sketched in Figure 1 [2,3,4,5,6,7,8,9,10,11,12,13]. Moreover, uranium is one of the few naturally occurring actinides [1], whereas the other members, with the exception of thorium, are considered to be human-made, despite their natural occurrence in traces as a result of uranium and thorium spontaneous fissions. In nature, uranium occurs as three main isotopes, where the non-fissile ${}^{238}$U is the most abundant, encompassing over 99% of the available uranium resources.

In the Earth’s crust, there are different types of uranium ores, such as uraninite or pitchblende (*x*UO${}_{2}\cdot y$UO${}_{3}$, with *y*/*x* < 2), carnotite (K${}_{2}$O·2UO${}_{3}\cdot$V${}_{2}$O${}_{5}\cdot x$H${}_{2}$O) and autunite (CaO·2UO${}_{3}\cdot$P${}_{2}$O${}_{5}\cdot x$H${}_{2}$O) [14,15]. In order to extract uranium from its ore, chemical processes utilizing acids and oxidizing agents are employed. Often, specific isotopic separation processes have to be applied in order to acquire the desired concentration of a given uranium isotope for energy generation purposes [1]. The most used isotopic separation process involves uranium conversion into the gaseous UF${}_{6}$ followed by isotope separation due to mass differences [16].

A number of reviews have been previously published covering the industrial use of uranium, i.e., in energy supply and related fields [15,17,18,19,20,21]. Organouranium compounds, i.e., the uranium equivalent of organometallic complexes, have attracted considerable attention, including reviews on ($\eta^{8}$-C${}_{8}$H${}_{8}$)${}_{2}$U and actinides cyclobutadienyl complexes, respectively [22,23].

Uranium coordination chemistry, in general, has been being widely investigated [9,24,25,26], especially with regard to possible applications in nuclear waste treatment [27], where the efficient extraction of uranium from the waste generated during the nuclear fuel cycle (so-called spent nuclear fuel), and its recycling into the nuclear fuel process is an important step to reduce long-term radioactive waste [28,29,30,31,32]. Uranium oxides have been used for a long time as nuclear fuel [33,34]. However, their handling in the spent nuclear fuel regeneration process is difficult [35].

N-substituted uranium amides, including uranyl nitrates with picolinamide-based ligands, uranyl di-amide complexes and uranyl Schiff-base complexes, have been found to be highly promising alternatives. They are fully combustable, show high hydrolytic and radiation stability and produce degradable products [11,36,37,38]. Furthermore, N-substituted uranium amides show diverse coordination behavior. They can bind through the carbonyl oxygen atom with the central metal atom or ion, and they can form bonds with a metal ion through the deprotonated N atom of the ligands.

It has been suggested that, in both cases, bonding can be strengthened in the presence of a preferable anchoring group that can form a five or six-membered chelate ring, as in picolinamide, which is an interesting aspect waiting to be further explored [11]. Some attempts have been made to estimate the uranium-oxygen (UO) and uranium-nitrogen (UN) bond strengths in these compounds via geometrical parameters, atomic charges or normal vibrational frequencies; however, a quantitative measure of the UO and UN bond strength allowing the strategic modulation of these uranyl complexes is missing, a problem that we will tackle in the last part of this review via a new tool for quantifying the strength of chemical bonds and/or weak chemical interactions based on vibrational spectroscopy [39].

Although the use of uranium amides appears to be an attractive alternative pathway, there is still the general concern about the effects of nuclear energy use on nature and human health, as documented in a number of reviews, such as [9,40,41,42,43].

Uuranium can adopt different oxidation states ranging from III to VI, thus, leading to a rich repertoire of uranium-ligand (UL) bonding with diverse covalency and 5f-/6d-orbital contributions, including δ and ϕ back-donation [44]. While, e.g., charge- and electron-loaded ligands favor U(VI), U(III) is mostly found in the presence of strong donor ligands engaging in ionic, lanthanide-like bonding [13,45,46,47]. Uranium multiple bonding has been discussed [48,49] and exhibited new opportunities in small-molecule activation [10].

In particular, uranium nitride linkages with a formal UN triple bond have attracted attention [38]. Recently, uranium mononitride was suggested as an attractive fuel for a range of reactors [50]. However, in order to explore these possibilities, there is a quest for further research to fully understand the nature of the UN triple bond. A powerful tool for the investigation of electronic structure and bonding in uranium compounds, the strategic investigation and fine-tuning of uranium-based catalysts and the discovery of novel and unusual U-catalytic transformations is provided by computational chemistry [51], which can also be helpful for the analysis of available experimental IR/Raman spectra of uranium complexes to identify normal mode vibrations with a high percentage of UL character [11,12,37,52].

However, the theoretical description of uranium complexes requires a careful treatment of relativistic effects [53,54], which make sizable or even dominate contributions to the molecular properties of uranium compounds—in particular, to UL bonding. In general, the *s* electrons of relativistic atoms are contracted as a direct consequence of their higher attraction to the heavy nucleus. This is also experienced by *p* orbitals, though in a smaller amount; *d* and *f* orbitals become shielded by the inner *s* and *p*.

As a consequence, outer electrons are less attracted to the nucleus and more available for chemical bonding [1,55], resulting in a larger variety of oxidation states, diverse coordination compounds [1] and a number of special properties, such as the color of gold [56,57].

In order to capture the relativistic effects, one has to go beyond the non-relativistic Schrödinger equation and instead refer to the far more complex Dirac Equation [58], thus, leading to relativistic Dirac-exact wave functions and associated molecular properties [59,60,61]. The most rigorous way to include relativity in the calculation of molecular systems is to use Dirac’s full four-component (4c) formalism, leading to wave functions as vectors of four complex numbers (known as bispinors) [62].

A large variety of approximate methods have been derived over the years [39,63], motivated by the assumption that the full 4c approach is computationally too demanding and cannot be applied to larger molecules [64,65,66,67,68,69,70,71]. Quasi-four component and two-component forms [63] have been developed to make these calculations more practical, such as the Douglas–Kroll–Hess approach [72,73], Regular Approximations [74] and the Normalized Elimination of the Small Component (NESC) [75].

Over the years, our group has developed a powerful two-component (2c)-NESC method [55,76] that allows the accurate calculation of first- and second-order response properties, including vibrational frequencies [77,78,79,80,81,82,83,84], for a variety of density functional and correlation-corrected wave function methods, thus, guaranteeing a broad application range [55,85].

A recent review on this topic, in particular with regard to the description of actinides can be found in reference [39], which also addresses the popular use of effective core potentials to describe relativistic effects using a non-relativistic platform, which are a cheap alternative to complex Dirac-based methods [86]. However, these seriously fail in the case of uranium, leading to calculated molecular properties with errors as large as 20% [87]. Recently, we adopted a new version of NESC using the atomic approximation (NESCau), which substantially increases the efficiency of relativistic calculations [39,64] even for larger systems.

The review is structured in the following way. In Section 2, we focus on the use of uranium as nuclear fuel, followed by brief summary of the uranium cycle. In Section 3, case studies of human and environmental injuries caused by accidents and disasters occurred in power plants are presented. Section 4 approaches the main challenges with respect to nuclear waste treatment, targeting recent suggestions. Finally, in Section 5, we discuss uranium nitrides and their potential as novel nuclear fuels. We present a new effective way to quantitatively determine the strength of a chemical bond based on the local vibrational mode analysis (LMA), originally suggested by Konkoli and Cremer [88,89,90,91,92], and apply LMA to tackle the currently controversy about the U-O and U-N bond strengths in uranium nitrides. In Section 6, a final outlook is given.

## 2. Role of Uranium in the Nuclear Fuel Cycle

Uranium is mainly used in the nuclear fuel cycle sketched in Figure 2, an industrial process involving various activities to produce electricity from uranium in nuclear power reactors, i.e., the progression of nuclear fuel from creation to disposal. Typically, uranium is employed in the form of UO${}_{2}$. During uranium decay, fissile plutonium is produced. As spent fuel can be reprocessed and recycled, this explains the presence of a plutonium admixture with uranium in the subsequent fuel cycle [14,93].

Uranium is a relatively common element that is found throughout the world and mined in a number of countries. After mining, uranium ore is crushed in a mill and grounded to a fine slurry, which is leached in sulfuric acid (or sometimes a strong alkaline solution) to allow the separation of uranium from the waste rock. It is then recovered from solution and precipitated as uranium oxide concentrate, mostly U${}_{3}$O${}_{8}$, which is sometimes referred to as *yellowcake* uranium (caused by its khaki color). U${}_{3}$O${}_{8}$ concentrate typically contains more than 80% uranium. The original ore, by comparison, may contain as little as 0.1% uranium [14,93].

Often, the yellowcake needs to be highly purified, due to impurities originated from the ore that remain, a process that is highly specific and depends on the ore quality [94]. Alternate processes have been proposed, such as supercritical carbon dioxide extraction, in particular for phosphate ores [95,96]. Purified U${}_{3}$O${}_{8}$ is the product that is sold. About 200 tonnes are required to keep a large (1000 MWe) nuclear power reactor generating electricity for a year.

The uranium oxide obtained after uranium milling is not directly usable as fuel for a nuclear reactor—additional processing is required. Only 0.7% of natural uranium is fissile—capable of undergoing fission, the process by which energy is produced in a nuclear reactor. The form (or isotope) of uranium that is fissile is ${}^{235}$U, whereas the remainder is ${}^{238}$U. About 3.5–5% of the fissile isotope ${}^{235}$U is required to maintain the chain reaction. Therefore, a physical process commonly called *uranium enrichment* has to be invoked to increase (or ‘enrich’) the ${}^{235}$U concentration, which requires uranium to be in a gaseous form [1]. U${}_{3}$O${}_{8}$ is first refined to uranium dioxide UO${}_{2}$, which can be directly used as fuel for those types of reactors that do not require enriched uranium. For reactors requiring enriched uranium, UO${}_{2}$ is converted into uranium hexafluoride, which is a gas at relatively low temperatures. In order to produce UF${}_{6}$, UO${}_{2}$ is first converted into the solid UF${}_{4}$, followed by oxidation with F${}_{2}$, yielding UF${}_{6}$(g).

This route is often preferable instead of the direct conversion of the oxide into the fluoride, due to the lower fluorine demand. Solid state reactions often lead to byproducts, which also have to be taken into account [94,97,98,99]. A gas-phase mechanism for such reactions was previously studied theoretically [100]. Currently, centrifuges are used for U-enrichment processes. Nonetheless, there are alternative techniques, such as gaseous diffusion, which was widely used in the past, and laser-based processes, which are still in development [101].

Subsequently, uranium has to be re-converted to its oxide form. There are several processes available, including wet conversions employing aqueous ammonia, for example, which hydrolyses UF${}_{6}$, followed by calcination. An alternative process is dry conversion, which is known to be less injurious for the environment. Independent of the method used, it is important that the final product—the actual reactor fuel—has the correct oxygen to uranium ratio [93]. In order to increase the stability and strength, nuclear fuel can be alloyed with zirconium [14,93,102], which has also drawn attention due to its potential use in spent fuel reprocessing [103,104].

The fission process transforms U (and Pu) into lighter elements, denoted as *fissile products*, which are often also radioactive. Thus, there are safety concerns regarding nuclear waste. Spent fuel still contains large amounts of uranium, known as depleted uranium and, therefore, can be recycled and reused from nuclear waste [14]; however, the often highly radioactive fissile products have to be safely stored [14,105].

As alternative to uranium oxides, uranium nitrides are discussed as novel nuclear fuel as they display higher values for thermal conductivity and melting point [106]. The species N≡U≡N was first obtained in an argon matrix through a molecular nitrogen reaction with atomic uranium. Clearly, uranium has the potential to break the triple bond in N${}_{2}$ [107]. A recent investigation of N≡U≡N in both neutral and negatively charged states has led to new insights into the properties of this molecule in both the ground and excited states [108].

Other uranium nitrides followed, such as the related NUO, that was first observed in solid argon as a product of the reaction between U and NO, and density functional theory calculations predicted a duplet spin state [109]. NUO was also synthesized in solid neon by Zhou and Andrews [110]. A related species, NUNH, was observed as a product of the N≡U≡N reaction with hydrogen atoms, suggesting that NUNH features both double and triple uranium-nitrogen bonds [111].

## 3. Spent Fuel Recycling

The spent fuel contains uranium (about 96%), plutonium (about 1%) and high level radioactive waste products (about 3%). In order to recycle the spent fuel, uranium and plutonium have to be extracted from the nuclear waste. Current reprocessing plants dissolve the spent fuel and chemically separate it into those three components: uranium, plutonium and high level waste in most cases by dissolution in concentrated nitric acid, followed by extraction with tributyl phosphate (TBP) in kerosene. This leads to the complexation of uranium and plutonium, whereas the other fission products remain in the aqueous phase [112,113,114].

TBP, in turn, may undergo degradation, thus, requiring the removal of potential degradation products [94]. Therefore, alternative solvents have been proposed and successfully applied, such as Aliquat 336 [115] and Alamine 336 [116]. There are different ways to extract the left-over fission products. Alkali and alkaline earth metals can be removed through pH adjustments, followed by ion exchange [94]. The separation of lanthanides and actinides contained in spent fuel is a challenge for the nuclear industry due to their chemical similarities [117,118,119].

A number of complexation methods using different ligands and coordination patterns have been suggested [27]. In general, coordination chemistry has been successfully applied to extract depleted uranium from waste water for re-use. Apart from that, uranium complexes have shown to be potentially useful in catalysis [9,120]. Therefore, in the next section, the coordination chemistry of uranium will be further elucidated.

## 4. Coordination Chemistry of Uranium

The coordination chemistry of uranium dates back to at least the early 1800s. It has become an increasingly popular topic in the last 15–20 years, mostly because of the development of easy-to-synthesize, stable starting materials for, e.g., organometallic synthesis and catalysis. There is tremendous diversity in the type of ligands that have been found to form stable complexes with uranium [121]. Uranium atoms can adopt different oxidation states ranging from III to VI, leading to a rich repertoire of UL bonding with diverse covalency and 5f-/6d-orbital contributions. While, e.g., charge- and electron-loaded ligands favor U(VI), U(III) is mostly found in the presence of strong donor ligands engaging in ionic, lanthanide-like bonding [13,45,46,47].

The majority of uranium coordination compounds display either IV or VI oxidation states. In general, U(IV) molecules are more difficult to synthesize, as specific experimental conditions are required [122,123]. U(V) compounds tend to disproportionate into IV and VI oxidation states [14,99]. Nonetheless, procedures have been suggested for how to stabilize U(V) molecules while taking into consideration that the apparent U(V) instability could be a result of experimental limitations [124].

Uranium complexes with a UO${}_{2}$ group and are often called *uranyl compounds* [99]. High-level coupled-cluster calculations have predicted that UO${}_{2}^{+}$ hydrolysis leads to an intermediate adduct and UO(OH)${}_{2}^{+}$ as the final product (see Figure 3, molecule **1**). The corresponding U(VI) complex, i.e., UO${}_{2}^{2+}$ undergoes a similar reaction, as experimentally confirmed; however, it was found that the reaction barrier is slightly higher than that of the corresponding U(V) complex [125,126]. Many other reactions involving the uranyl cation have been studied thus far [127].

For some time, it was believed that the lowest oxidation of uranium was III [1,128]. However, in 2013, the first stable U(II) complex was synthesized in the form of [Cp${}_{3}^{\prime }$U]${}^{-}$, in which Cp${}^{\prime }$ accounts for the group C${}_{5}$H${}_{3}$(SiMe${}_{3}$) [129] (see Figure 3, complex **2a**). After that, other stable U(II) molecules were isolated, e.g., using sodium as a reducing agent in the synthesis of U(II) compounds [128]. In addition to U(IV) and U(III) metallocenes with a bent structure [3,5], recently, the first stable U(II) [$\eta {}^{5}$-C${}_{5}^{\;i}$Pr${}_{5}$)${}_{2}$U] sandwich complex (see Figure 3, complex **2b**) was synthesized in the reduction of U(III) by potassium graphite, and this has an unusual linear Cp-U-Cp angle [4].

Electronic structure calculations exhibit a mixing between 6d${}_{z^{2}}$ and 7s orbitals, which, according to the authors, is the main reason for the linear arrangement found in this complex. Recently, our group conducted a NESCau study on actinides sandwich complexes, including uranium; an example can be found in Figure 3, complex **3** [39]. A theoretical study on a set of ($\eta^{5}$-C${}_{5}^{\;i}$Pr${}_{5}$)${}_{2}$U-derived complexes is currently under investigation in our group.

It has been hypothesized that bulky substituents are necessary to isolate this stable U(II) uranocene with substantial covalent U-π character, caused by uranium 5f and 6d orbital participation. Negatively charged U(II) metallocenes have been widely investigated computationally [130]. Confirmed by both experiment and theory, the (SiMe${}_{3}$) group can serve as a stabilizing substituent due to its electron withdrawing character; however, stable U(II) complexes with electron-donating ligands have been reported as well, such as [Cp${}^{\prime }($C${}_{5}$Me${}_{4}$H)${}_{3}$U]${}^{-}$ [131]. According to these studies, the high-spin quintet state is the most stable in these U(II) compounds, where uranium adopts a valence configuration of 5f${}^{3}$6d${}^{1}$.

Thus far, no U(I) complexes have been observed, except a divalent uranium, which behaves in 2- or 3-electron reductions, such as a U(I) species and, therefore, was named U(I) *synthon* [132] (see Figure 3, complex **4**). As discussed above, N-substituted uranium amide complexes (see Figure 3; complex **5a**: uranyl nitrates, including picolinamide-based ligands [11]; complex **5b**: uranyl diamide complexes [36]; complex **5c**: uranyl Schiff-base coordination complexes [37]) have been recently discussed as attractive candidates for the next generation of reactors with regard to both the actual nuclear fuel cycle, spent fuel reprocessing and recycling [11,36,37], as well as complexes with uranium multiple bonding, including uranium nitrides [10,48,49,50] (see Figure 3, complex **6**).

The [UN${}_{2}$] isoelectronic nitrogen analogue of the well know uranyl cation [UO${}_{2}$]${}^{2+}$ [127] has been suggested as future player for ceramic fuel in nuclear reactors [48,108]. Recent X-ray structures of the triuranium complexes, isolated in an argon matrix, supported by Raman spectra suggest a central U(VI) [UN${}_{2}$] core with U≡N triple bonding [48]. Photoelectron spectroscopy predicts that electron attachment and electronic excitation significantly bends the [UN${}_{2}$] unit and elongates the U≡N bond [108]. Since excited-state uranium nitrides may be prevalent in nuclear reactors, it will be interesting to systematically investigate the strength of the interaction between the pentagonal bipyramidal [UN${}_{2}$R${}_{2}$R’${}_{3}$] core and the two flanking uranium cations as well as their influence on the strength of the U≡N triple bond in both the ground and excited state.

## 5. Uranium and Human Health

Despite all safety efforts, disasters involving nuclear power plants have occurred, and serious consequences have arisen [133,134,135]. The Chernobyl disaster, which took place in the Ukraine in 1986, is an appalling example. A fire that broke out during a safety test led to an explosion and the release of radioactive material into the environment with catastrophic human and nature impacts [136]. Melted material (so-called *lava*) is a signature of such an accident, causing the spread of radio-nuclides and an emerging risk of secondary explosions [134,135].

People in the surrounding area (workers and inhabitants) were exposed to high levels of radiation, and an increased risk of cancer, mainly leukemia, was observed [137], as well as congenital malformations, as a consequence of the exposure to radiation [138]. In 2011, following an Earthquake, a horrible accident occurred at Fukushima Dai-ichi Nuclear Power Plant in Japan [133,139]. Although the power plant was immediately shut down after the earthquake, the following tsunami led to power outages, which interrupted the reactor cooling system [135]. Similarly to Chernobyl, nuclear meltdowns and radiation leakage arose. The Fukushima disaster also caused the release of contaminated material into the ocean as a direct consequence of the flooding; a portion of the contaminated seawater was later recycled into the environment [133,135].

In both cases, many people lost their lives, and regions close to the plant had to be evacuated and remain inhabitable up to date [140]. Both accidents initiated vivid debates on nuclear energy, including the problem of how to best store nuclear waste on a long-term basis and how to make these plants more efficient while maintaining all possible safety standards. A serious health problem is connected with *depleted Uranium* (DU), a byproduct of natural uranium after the enrichment and extraction of ${}^{235}$U that is used to provide fuel for nuclear power plants and/or to charge nuclear weapons [141,142].

Aerosol inhalation is considered the primary route of DU exposure. Although laboratory tests have confirmed that inhalation of DU aerosol can cause lung, kidney and other organ damage, epidemiological studies have found no conclusive evidence that persons in areas with prolonged exposure to DU-containing bombs are affected. Apart from that, DU can induce multiple other health effects, such as renal tubular necrosis and bone malignancies [143].

Uranium contamination is a global health concern. Regarding natural or anthropogenic uranium contamination, the major sources of concern are groundwater, mining, phosphate fertilizers, nuclear facilities and military activities. Apart from radioactivity, uranium shows the typical chemotoxicity of a heavy metal [144]. The consumption of water or food contaminated with uranium can lead to chronic kidney diseases and failure [145,146].

According to the World Health Organization (WHO), uranium intake from food or air is relatively low and, according to the region’s soil, water consumption might be a source of contamination. A guideline of 30 mg/L has been established; however, thus far, there is not sufficient evidence regarding cancer risks and other health issues [147]. Nonetheless, despite its lower overall chemical toxicity, as discussed above, a number of uranium isotopes are highly radioactive, including γ ray emissions. These isotopes are highly carcinogenic and dangerous for humans and wildlife.

In nuclear medicine, radionuclides are widely used in both diagnostics and treatment. Uranium is not directly employed for this to date; however, it is used to provide radioisotopes to be used in nuclear medicine. The most known example is ${}^{99}$Mo/${}^{99m}$Tc, which is used in imaging diagnostics. Another example of a radionuclide obtained from uranium is ${}^{229}$Th/${}^{225}$Ac. It is used in *Targeted Alpha Therapy*, a class of radionuclide therapy [148,149]. ${}^{235}$U is used to produce ${}^{99}$Mo through a fission reaction.

Its daughter ${}^{99m}$Tc is widely used in diagnostics. The majority of ${}^{99}$Mo/${}^{99m}$Tc is produced in research reactors; nonetheless, there are other routes, such as cyclotrons, which have the disadvantage of a lower production yield [150]. ${}^{229}$Th results from ${}^{233}$U decay. Its decay leads to ${}^{225}$Ac, which has been used in the treatment of several types of cancer, including prostate cancer. A high level of purity is needed due to the fact that radionuclide impurities may be potentially toxic [149].

## 6. UN Bond Strength in Uranium Nitrides

Given the increasing interest in uranium nitrides, in this section, we introduce an efficient tool to quantitatively assess the strength of a chemical bond/weak chemical interaction based on vibrational spectroscopy and showcase its application for some selected uranium nitrides, N≡U≡N, U≡N, N≡U=NH and N≡U=O, clarifying if these molecules exhibit a UN triple bond, as hypothesized in the literature.

LMA has been extensively described in a recent review [151]; therefore, in the following, only highlights are summarized. The idea of characterizing a chemical bond via the normal mode stretching force constant has its roots in the 1920s and 1930s and is based on the famous *Badger rule*, an inverse power relationship between the bond length and stretching force constant [152]. While this rule works perfectly fine for diatomic molecules, its extension to polyatomic molecules has been a major obstacle [151,153] because normal vibrational modes tend to delocalize over the molecule rather than being localized in a specific bond [154,155].

As of that important fact, the related normal mode stretching force constants cannot serve as a suitable bond strength measure. LMA, originally suggested by Konkoli and Cremer [88,89,90,91,92], solved this problem, which led to a new measure of the intrinsic strength of a chemical bond or weak chemical interactions and a generalized Badger rule [153] based on local mode force constants $k^{a}$. A comprehensive discussion of the underlying theory of LMA, following two independent routes to derive local vibrational modes, is given in Ref. [151].

Local mode force constants, contrary to normal mode force constants are independent of the choice of the coordinates used to describe the molecule in question [153,156,157]. They are sensitive to differences in the electronic structure (e.g., caused by changing a substituent), and because they are, in contrast to frequencies, independent of the atomic masses, they capture pure electronic effects. In their landmark paper, Zou and Cremer [158] proved that the local stretching force constant $k_{n}^{a}$(AB) reflects the intrinsic strength of the bond/interaction between two atoms A and B, which is described by an internal coordinate $q_{n}$.

In essence, LMA has advanced as a powerful analytical tool, extensively applied to a broad range of chemical systems from simple molecular systems to systems in solution [159,160] to proteins [161,162], accounting for both covalent bonds [153,158,163,164,165,166,167,168,169,170,171,172,173,174,175] and non-covalent interactions [172,176,177,178,179,180,181,182,183,184,185,186,187,188,189,190], including hydrogen bonds [191,192,193,194,195,196,197,198,199,200,201]. Recently, LMA theory has been extended to periodic systems, a large step forward to the quantitative description of bonding in crystals and solids [190,202,203].

For the comparison of larger sets of $k^{a}$ values, the use of a relative bond strength order (BSO) is more convenient. According to the generalized Badger rule derived by Cremer, Kraka and co-workers [153,169], both are connected via a power relationship of the form
(1)$$BSO=u{\left(k^{a}\right)}^{v}$$
where the parameters *u* and *v* are obtained from two reference molecules with known BSO and $k^{a}$ values and the request that, for a zero force constant, the BSO value is also zero. For example, for CC bonds, suitable references are ethane and ethylene with bond orders *n* = 1 and *n* = 2, respectively, [163,174,204]. In the case of more complex bonding situations, e.g., metal–ligand bonding, guidance by Mayer bond orders [205,206,207] can be utilized [208]. It is important to note that the reference molecules have to be calculated with the same model chemistry as the set of molecules under investigation.

In this work, H${}_{2}$N-UH and HN=UH${}_{2}$ were used as reference molecules for uranium-nitrogen single and double bonds with $k^{a}$ values of 2.253 and 4.865 mDyn/Å, respectively. Optimized UN bond lengths of 2.166 and 1.872 Å, respectively, obtained in this work are in excellent agreement with previously reported bond lengths [209]. The corresponding Mayer bond orders for the UN single and double bonds were 1.150 and 2.265. Scaling the Mayer bond order of the UN single bond to the value 1.0, the corresponding bond order of the UN double bond is 1.97, which led to *u* and *v* values of 0.4894 and 0.8803, respectively, as used in this work.

For the uranium-oxygen bonds, uranyl hydroxide, UO${}_{2}$(OH)${}_{2}$ (complex **1** in Figure 3) was used as a single reference molecule, because it has both single and double uranium-oxygen bonds. We obtained $k^{a}$ values for the UO single and double bonds of 2.791 and 7.114 mDyn/Å, respectively, and optimized UO bond lengths of 2.100 and 1.762 Å, respectively. The corresponding Mayer bond orders for the UO single and double bonds were 1.175 and 2.270. Scaling the Mayer bond order of the UO single bond to the value 1.0, the corresponding bond order of the UO double bond is 1.932, which led to *u* and *v* values of 0.4855 and 0.7041, respectively, as used in this work.

The quantum theory of atoms-in-molecules (QTAIM), developed by Bader [210,211,212,213], identifies, analyzes and characterizes chemical bonds via the topological features of the total electron density ρ(r). In this work, we used QTAIM as a complementary tool to LMA to determine the covalent character of the UN and UO bonds via the Cremer–Kraka criterion [214,215,216] of covalent bonding. The Cremer–Kraka criterion is composed of two conditions: (i) a necessary condition: the existence of a bond path and bond critical point **r**${}_{b}$, i.e., the (3,−1) saddle point of electron density ρ(r) between the two atoms under consideration; and (ii) a sufficient condition: the energy density *H*(**r**${}_{b}$) at that point is smaller than zero. *H*(**r**) is defined as:(2)H(r)=G(r)+V(r)
where *G*(**r**) is the kinetic energy density and *V*(**r**) is the potential energy density. The negative *V*(**r**) corresponds to a stabilizing accumulation of density whereas the positive *G*(**r**) corresponds to depletion of the electron density [215]. As a result, the sign of *H*(**r**${}_{b}$) indicates which term is dominant [216]. If *H*(**r**${}_{b}$) < 0, the interaction is considered to be covalent in nature, whereas *H*(**r**${}_{b}$) > 0 is indicative of electrostatic interactions.

All geometry and frequency calculations were performed in the is work with the PBE0 density functional [217,218,219] combined with the cc-pwCVTZ-X2C basis set for uranium and the corresponding cc-pVTZ basis set for the non-relativistic atoms [220,221,222,223,224]. Relativistic effects were included through the NESCau Hamiltonian [64] (recently applied to a larger set of actinide sandwich complexes [39]) using the COLOGNE2020 program package [225]. LMA properties were calculated with the LmodeA program [226], and the energy density at the bond critical points was obtained using the AIMALL software [227].

In Table 1, LMA and QTAIM results for N≡U≡N, U≡N, N≡U=NH, N≡U=O and UO${}_{2}$ are summarized. Figure 4 shows the BSO values obtained from the local stretching force constants $k^{a}$, Figure 5 correlates *H*(**r**${}_{b}$) and $k^{a}$. The BSO values (see Table 1 and Figure 4) clearly show that, in all four compounds, U≡N bonds are of triple bond character, thus quantifying, for the first time, previous speculations and suggestions [109,110,111]. The lowest value of 2.976 was found for U≡N, comparable values of 3.081 and 3.087 for N≡U≡N and N≡U=NH, respectively, and the strongest triple bond for N≡U=O with BSO = 3.238.

The U=NH bond in U≡N, N≡U=NH can be identified as double bond with BSO = 1.989, and the U=O bond in N≡U=O and UO${}_{2}$ is a slightly stronger double bond with BSO = 2.066 and 1.966, respectively. The U=O double bond of the UO${}_{2}$ triplet ${}^{3}\Phi_{u}$ ground state mirrors with a bond length of r = 1.775 Å and BSO of 1.966 the U=O bond in N≡U=O (r = 1.761 Å, BSO = 2.066), whereas the U=O bond in the excited UO${}_{2}$${}^{1}\Phi_{u}$ singlet state is considerably longer and weaker (r = 1.836 Å, BSO = 1.689).

As is depicted in Table 1 and Figure 5, all UN and UO bonds investigated are of covalent character showing the general trend that a stronger bond is also more covalent. There is no perfect correlation between these two properties, which is the result of the fact that whereas local force constants reflect the electronic environment, the energy density is taken at only one point on the bond path [39].

## 7. Conclusions and Future Perspectives

This review summarizes the recent research on the use of uranium as nuclear fuel, including recycling and health aspects elucidated from a chemical point of view, i.e., emphasizing the rich uranium coordination chemistry, which has also raised interest in using uranium compounds in synthesis and catalysis. A number of novel uranium coordination features are addressed, such as the emerging number of U(II) complexes and uranium nitrides as a promising class of materials for more efficient and safer nuclear fuels. The current discussion about uranium triple bonds was addressed by quantum chemical investigations using local vibrational mode force constants as a quantitative bond strength descriptor based on vibrational spectroscopy.

The local mode analysis for selected uranium nitrides, N≡U≡N, U≡N, N≡U=NH and N≡U=O, could confirm and quantify, for the first time, that these molecules exhibit a UN triple bond as has been hypothesized in the literature. After this proof of concept, we are currently expanding our investigation to a larger series of molecules with potential UN triple bonds.

Future work in our group will also include (i) a comprehensive study of the intrinsic strength of UL bonds—in particular, their covalent or non-covalent character as well as the role of the f orbitals for UL bonding, including δ and ϕ back-donation [44], which is a necessary prerequisite for the strategic fine-tuning of uranium-based catalysts and the discovery of novel and unusual U-catalytic transformations; (ii) a systematic analysis of available experimental IR/Raman spectra of uranium complexes with LMA to identify normal mode vibrations with a high percentage of UL character [11,12,37,52]; (iii) following the procedure determined for actinide sandwich compounds [39], we will derive valuable guidelines for the fine-tuning of these U(II) sandwich compounds in their role as powerful reducing agents, including the evaluation of current suggestions that bulky substituents are necessary to isolate this stable U(II) uranocene with substantial covalent U-π character, caused by uranium 5f and 6d orbital participation; and (iv) the investigation of a series of uranyl nitrates with and without picolinamide-based ligands, uranyl diamide and uranyl Schiff-base coordination complexes and the compilation of a comprehensive list of U-O and U-N local force constants together with electron density data quantifying the covalent character of these bonds.

Based on our results, we will determine a design protocol to increase the U-ligand bond strength towards an improved remediation of nuclear waste from reactor fuels. We hope that this review will inspire the community interested in uranium chemistry and will serve as an incubator for fruitful collaborations between theory and experimentation to explore the wealth of uranium chemistry.

## Figures and Tables

**Figure 1 ijms-23-04655-f001:**
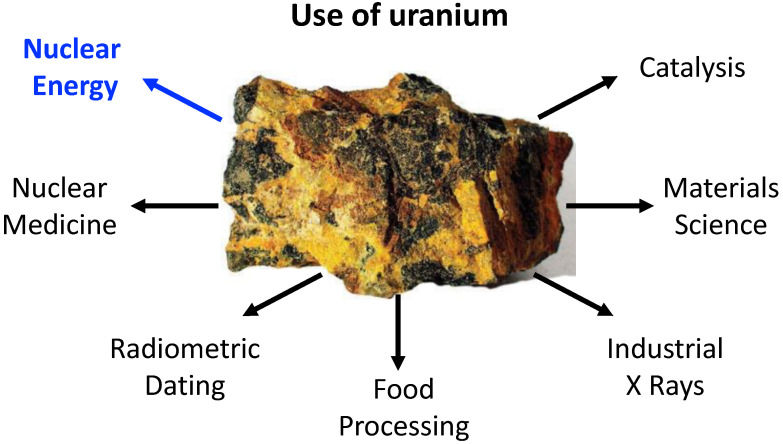
Examples of the broad range of uranium use.

**Figure 2 ijms-23-04655-f002:**
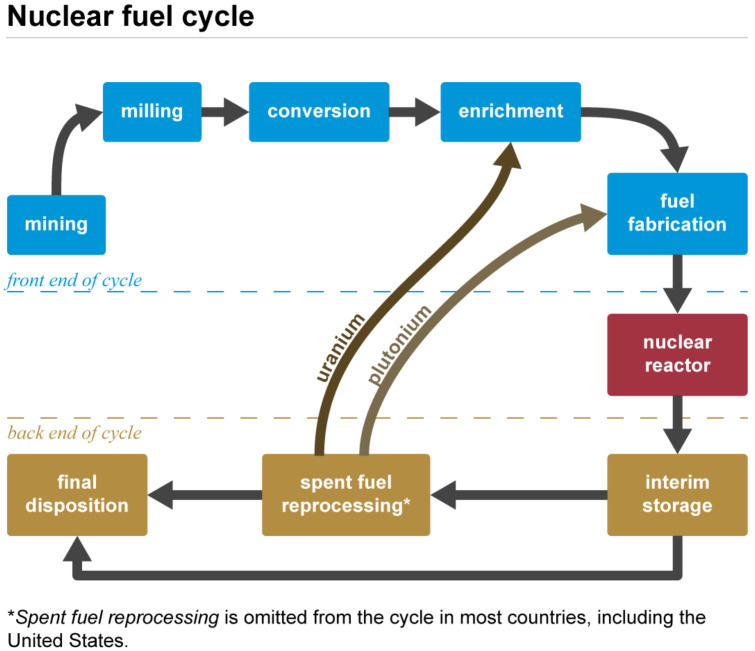
Nuclear fuel cycle. Source: Pennsylvania State University Radiation Science and Engineering Center (public domain).

**Figure 3 ijms-23-04655-f003:**
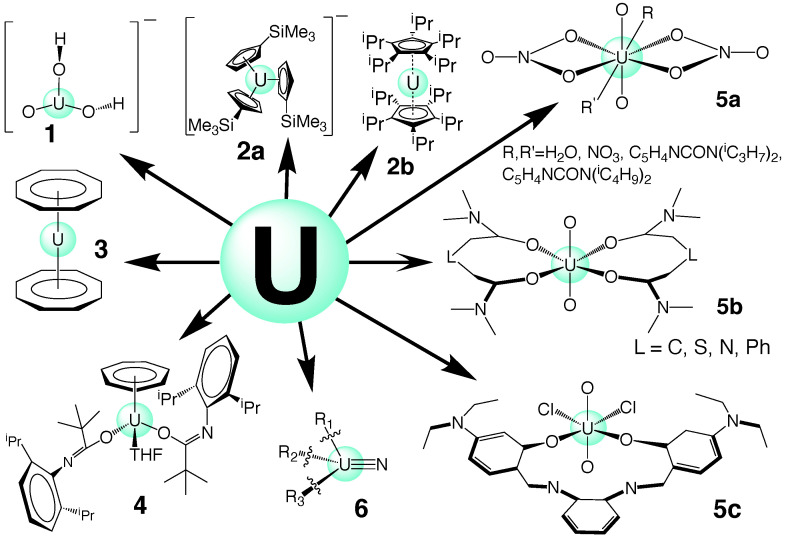
Representative uranium complexes; the numbers refer to the discussion in the text.

**Figure 4 ijms-23-04655-f004:**
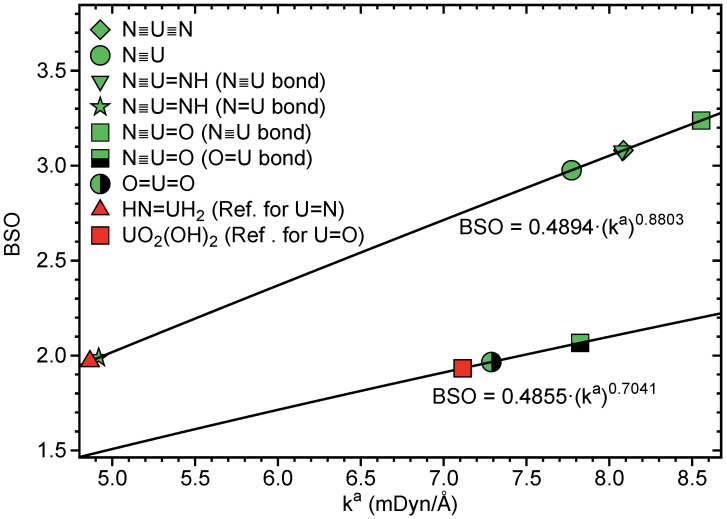
BSO values of UN bonds for the selected uranium nitrides derived from *k*${}^{a}$ values via the power relationship described above. NESCau/PBE0//cc-pwCVTZ-X2C (uranium) and NESCau/PBE0//cc-pVTZ (N,O,H) level of theory.

**Figure 5 ijms-23-04655-f005:**
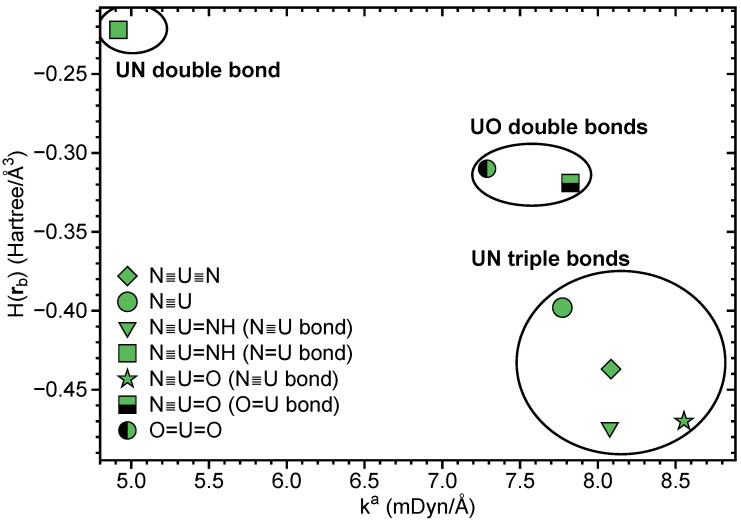
Correlation between H(**r**${}_{b}$) and $k^{a}$ for UN and UO bonds. NESCau/PBE0//cc-pwCVTZ-X2C (uranium) and NESCau/PBE0//cc-pVTZ (N,O,H) level of theory.

**Table 1 ijms-23-04655-t001:** Bond lengths in Å, *k*${}^{a}$ in mDyn/Å, BSO and H(**r**${}_{b}$), in Hartree/Å${}^{3}$, spin multiplicity and oxidation number of uranium for N≡U≡N, U≡N, N≡U=NH, N≡U=O and UO${}_{2}$${}^{\left(a\right)}$ calculated at the NESCau/PBE0//cc-pwCVTZ-X2C (uranium) and NESCau/PBE0//cc-pVTZ (N,O,H) level of theory.

	U≡N				U=L; L=N,O				U State
	r	*k* ${}^{a}$	BSO	H(r${}_{b}$)	r	*k* ${}^{a}$	BSO	H(r${}_{b}$)	
N≡U≡N	1.713	8.085	3.081	−0.437	-	-	-	-	${}^{1}\Sigma_{g}$—U(VI)
N≡U	1.732	7.772	2.976	−0.398	-	-	-	-	${}^{4}\Sigma$—U(III)
N≡U=NH	1.695	8.077	3.078	−0.474	1.828	4.918	1.989	−0.222	${}^{2}\Delta$—U(V)
N≡U=O	1.697	8.554	3.238	−0.470	1.761	7.824	2.066	−0.319	${}^{2}\Phi$—U(V)
UO${}_{2}$	-	-	-	-	1.836	5.883	1.689	−0.239	${}^{1}\Phi_{u}$—U(IV)
UO${}_{2}$	-	-	-	-	1.775	7.288	1.966	−0.310	${}^{3}\Phi_{u}$—U(IV)

^(*a*)^${}^{3}\Phi_{u}$ is the ground state of UO_2_ being 40.0 kcal·mol^−1^ more stable than the singlet ${}^{1}\Phi_{u}$ state.

## Data Availability

All data supporting the results of this work are presented in tables and figure of the manuscript.

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
