# Peer review of "Uranium: The Nuclear Fuel Cycle and Beyond"

_ijms, 2022, doi:10.3390/ijms23094655_

Round 1

Reviewer 1 Report

There are many papers and reviews about different aspects of a very wide scientific area such as uranium coordination chemistry and the nuclear fuel cycle.

In my opinion the review of Peluso and Kraka is extremely general and readers will not get a really deep information about the current state-of-art. Many papers of Profs. Peter C. Burns, Rodney Ewing, May Nyman, Alexandra Navrotsky, Omar Farha, Hans-Conrad zur Loye, Sergey V. Krivovichev, Christopher Cahil, and Thomas Albrecht-Schmitt about the chemistry of uranium have been ignored. 

I believe that the readers will be confused why authors decided to write such a brief review about such a wide field of science. I do not see any adequate ways to improve or expand the review because it looks like a section "Introduction" for the Students' Textbook about General aspects of Radiochemistry.

Author Response

Reviewer 1

There are many papers and reviews about different aspects of a very wide scientific area such as uranium coordination chemistry and the nuclear fuel cycle.
In my opinion the review of Peluso and Kraka is extremely general and readers will not get a really deep information about the current state-of-art. Many papers of Profs. Peter C. Burns, Rodney Ewing, May Nyman, Alexandra Navrotsky, Omar Farha, Hans-Conrad zur Loye, Sergey V. Krivovichev, Christopher Cahil, and Thomas Albrecht-Schmitt about the chemistry of uranium have been ignored.

I believe that the readers will be confused why authors decided to write such a brief review about such a wide field of science. I do not see any adequate ways to improve or expand the review because it looks like a section "Introduction" for the Students' Textbook about General aspects of Radiochemistry.
Following the suggestion of this reviewer, we included the suggested literature by Burns, Ewing, Navrotsky and Krivovichev on nuclear waste and the mineralogy/thermodynamics of uranium, respectively, see references 15, 20, and 21 of the revised manuscript. Burns review (reference 139) has been added to the discussion on nuclear disasters; see also lines 270-271, 276-281. zur Loye’s studies on U(IV) (references 125) and Ilton’s studies on pentavalent uranium (reference 126) have been included; see also lines 213 – 217.

Reviewer 2 Report

figure 4 and 5 should be reworked to be more readable. Particularly figure 4 has to provide indication of what the green points are representing.

At raw 339 revise "[146,151,156? ? –167]" deleting the question marks.

Author Response

Reviewer 2

figure 4 and 5 should be reworked to be more readable. Particularly figure 4 has to provide indication of what the green points are representing.
Following the suggestion of this reviewer, we modified Figures 4 and 5.

At raw 339 revise "[146,151,156? ? –167]" deleting the question marks.

Corrected.

Reviewer 3 Report

The article "Uranium: the Nuclear Fuel Cycle and Beyond" by Barbara Peluso and Elfi Kraka summarizes the latest developments in the use of uranium as a nuclear fuel, including aspects of utilization and health protection. The review also includes the latest knowledge of the author in terms of the description of uranium nitrides and the bond strength of similar compounds. The work looks quite elaborate and interesting to read.

Еhe number of references used in the work is sufficient and all references look enought relevant and justified.

As for the design of the work, I had questions about Figure 1 (page 2) - it looks too garish (in my opinion) and is poorly perceived by the reader in such a neat scientific work. I recommend making the drawing less "clipartistic", if possible.

In general, I didn't have a lot of questions about the details regarding the target content of this review article.

However, I have to draw the authors' attention to the text containing historical inaccuracy: on page 8, line 265 ", which took place in Ukraine in 1986" - Ukraine became an independent country on August 24, 1991, on the day of signing the Declaration of Independence of Ukraine. So if you want to write in which country the Chernobyl disaster occurred, you must specify the USSR.

I wish the authors success and progress in future research, as I am personally interested in the energy aspects in the structure of uranium compounds, and especially their interaction with the surface of carriers.

Author Response

Reviewer 3

The article "Uranium: the Nuclear Fuel Cycle and Beyond" by Barbara Peluso and Elfi Kraka summarizes the latest developments in the use of uranium as a nuclear fuel, including aspects of utilization and health protection. The review also includes the latest knowledge of the author in terms of the description of uranium nitrides and the bond strength of similar compounds. The work looks quite elaborate and interesting to read. The number of references used in the work is sufficient and all references look enought relevant and justified. We appreciate the kind comments of this reviewer.

As for the design of the work, I had questions about Figure 1 (page 2) - it looks too garish (in my opinion) and is poorly perceived by the reader in such a neat scientific work. I recommend making the drawing less "clipartistic", if possible.
Figure 1 has been replaced with a less clipartistic one.

In general, I didn't have a lot of questions about the details regarding the target content of this review article.